# Characterising the Gene Expression, Enzymatic Activity and Subcellular Localisation of *Arabidopsis thaliana* Metacaspase 5 (*AtMCA-IIb*)

**DOI:** 10.3390/biology12091155

**Published:** 2023-08-22

**Authors:** Zulfazli M. Sobri, Patrick Gallois

**Affiliations:** 1Department of Bioprocess Technology, Faculty of Biotechnology and Biomolecular Sciences, Universiti Putra Malaysia, Serdang 43400, Selangor, Malaysia; zulfazli@upm.edu.my; 2Bioprocessing and Biomanufacturing Research Centre, Faculty of Biotechnology and Biomolecular Sciences, Universiti Putra Malaysia, Serdang 43400, Selangor, Malaysia; 3Faculty of Biology, Medicine and Health, University of Manchester, Oxford Road, Manchester M13 9PL, UK

**Keywords:** ER-stress, metacaspase, plant proteases, protease activity, redundancy

## Abstract

**Simple Summary:**

Metacaspases are proteases present in bacteria, protists, algae, plants and fungi but not in animals. In a given organism, metacaspases are present in families with multiple members, making it difficult to determine which family member is responsible for what activity. One way to solve this issue is to search for specific cells or specific conditions where a single family member is active in the absence of other members. With this aim in mind, we report the characterisation of metacaspase number 5 from the plant *Arabidopsis*. It is a little-studied member of the metacaspase class that we detected only under very specific contexts, opening a window for the study of how it helps plants to cope with stress. Furthermore, we gathered important information on the property of the protein itself. The data presented provide additional insights into deciphering the function of metacaspases in plants and will support future research toward further understanding of their mode of action. This knowledge will ultimately inform the creation of climate-resilient crops to provide food security for a growing human population.

**Abstract:**

Metacaspases are a class of proteases found in plants that have gained attention in recent years due to their involvement in programmed cell death (PCD) and other essential cellular processes. Although structurally homologous to caspases found in animals, metacaspases have distinct properties and functions. There are nine metacaspase genes in the *Arabidopsis thaliana* genome; these can be type I or type II, and working out the function of each member of the gene family is challenging. In this study, we report the characterisation of one *Arabidopsis* type II metacaspase, metacaspase-5 (*AtMC5; AtMCA-IIb*). We detected the expression of *AtMC5* only under specific conditions with a strong upregulation by ER stress and oxidative stress at a narrow 6 h time point. Recombinant AtMC5 was successfully purified from *E. coli,* with the recombinant AtMC5 working optimally at pH 7, using an optimised reaction buffer containing 10 mM calcium chloride together with 15% sucrose. Like other metacaspases, AtMC5 cleaved after arginine residue and demonstrated a substrate preference towards VRPR. Additionally, *AtMC5-RFP* was shown to be localised in the cytosol and nucleus of transfected cells. We found no evidence of a strong link between AtMC5 and PCD, and the data provide additional insights into the function of metacaspases in plants and will aid in future research toward further understanding their mode of action.

## 1. Introduction

Plant metacaspases are a class of proteases that have gained considerable attention in recent years due to their involvement in programmed cell death (PCD) and other essential cellular processes in plants. These proteases are distantly related to the caspases found in animals but have distinct properties and functions. Vercammen et al. [1] were the first to report the substrate specificity of *AtMC4 (AtMCA-IIa)* and *AtMC9 (AtMCA-IIf*) that cleaved after arginine and lysine and not aspartic acid. Over the years, plant metacaspases have been found to have a role in PCD, either in development [2,3] or during biotic stress [4] and abiotic stress [5,6]. Besides PCD, metacaspases can modulate senescence [7]. At the cellular level, metacaspases carry out a range of functions, for example, in protein aggregate clearing [7] or substrate inactivation [8]. Moreover, metacaspases have been shown to be involved in peptide signalling during wounding or oxidative stress [9,10,11]. In *A. thaliana*, the type II metacaspase family comprises six members (*AtMC4/AtMCA-IIa, AtMC5/AtMCA-IIb, AtMC6/AtMCA-IIc, AtMC7/AtMCA-IId, AtMC8/AtMCA-IIe* and *AtMC9 (AtMCA-IIf*), which suggests some specialisation among the family members either by function or expression specificity, with *AtMC4, AtMC8* and *AtMC9* being the most studied members so far. While genetic redundancy may mask some of the metacaspase functions when studying single mutant phenotypes, some single metacaspases have been assigned to a role in specific contexts. For example, *AtMC4* regulates stem cell homeostasis in *Arabidopsis* by catalysing the cleavage of the AtLa1 protein in response to environmental stress [12]. *AtMC8* is involved in oxidative stress-induced programmed cell death (PCD) in *A. thaliana* [6] and as a positive mediator of cell death induced by fungal infection [5]. *AtMC9* was found to be expressed in the xylem of roots and shoots, and the knockout of *AtMC9* has a reduced corpse clearance during xylem PCD [2]. To further understand the function of metacaspases in plants, every member of the *Arabidopsis* metacaspase family needs to be characterised. Here we focus on *AtMC5*. The membrane-associated transcription factor *NAC089* was found to bind to the promoter of many downstream PCD-related genes, including *AtMC5*, and a knockout of *NAC089* did not express *AtMC5* upon endoplasmic reticulum (ER) stress treatment [13]. These findings suggest that *AtMC5* may play a role in ER stress-induced PCD and is regulated by *NAC089*. In this study, we aim to characterise the induction of *AtMC5* expression by different stresses and purify recombinant AtMC5 to define its enzymatic activity.

## 2. Material and Method

### 2.1. Plant Growth Condition

*A. thaliana* seeds were surface sterilised using 70% ethanol for 10 min and air-dried under a fume hood. The seeds were then resuspended in 0.2% agar (*w*/*v*) and sown onto Petri dishes containing half-strength MS media. Seeds were stratified for two days at 4 °C and incubated at 22 °C under an 8 h light and 16 h dark cycle unless otherwise stated. For ER stress treatment, 5 µg/mL tunicamycin was added into sterile half-MS media. For nitrogen starvation treatment, macronutrients in the absence of nitrogen (3 mM CaCl_2,_ 1.5 mM MgSO_4,_ 1.25 mM KH_2_PO_4,_ 5 mM KCl, 2 mM MES-KOH and 1% sucrose) were supplemented into MS micronutrient media (Sigma, St. Louis, MO, USA). For oxidative stress, 10 µM methyl viologen (Sigma) was added into sterile half-strength MS media. For senescence experiments, *A. thaliana* seeds were sown directly in compost soil and stratified for two days at 4 °C, and were grown up to 8 weeks in a growth cabinet at 22 °C under an 8 h light and 16 h dark cycle.

### 2.2. β-Glucuronidase (GUS) Histochemical Staining

Stratified *A. thaliana* seedlings were grown on half-strength MS plates for 10 days. The whole seedling was immersed in 90% acetone for tissue fixation, incubated at room temperature for 20 min, and then washed twice with phosphate buffer. Samples were then incubated in a staining buffer containing 1 mM of the substrate 5-bromo-4-chloro-3-indolyl β-D-glucuronide (X-gluc) and incubated at 37 °C for 4 to 16 h. Potassium ferri- and ferrocyanide (0.5 mM) were also added to the staining buffer to provide more precise localisation by preventing diffusion of the indoxyl intermediate. Samples were finally mounted on a glass slide in a chloral hydrate solution (2 mg/mL water) to extract chlorophyll. All seeds used for AtMC5 GUS expression were from a proAtMC5::AtMC5::GUS line (gift of Pr H. Tuominen, Uppsala, Sweden).

### 2.3. Sytox Green Staining

Ten-day-old *A. thaliana* roots were treated with 1 µM sytox green (Thermo Fisher Scientific, Loughborough, UK) for 10 min, followed by three washes with Milli Q water, before being mounted for fluorescence imaging.

### 2.4. DNA Sequencing

The DNA sequencing reactions were carried out using BigDye Terminator v3.1 (Applied Biosystems, Warrington, UK) and processed at the DNA Sequencing Facility at the University of Manchester.

### 2.5. Point Mutation

A single-point mutation was introduced using the QuikChange method. A primer pair—5′- CTCAGACTCTGCTCACAGTGGTGGTCTCATCC-3′ and 5′-GGATGAGACCACCACTGTGAGCAGAGTTCGAG -3′—was used to introduce a point mutation at position C139A, and full plasmid amplification was carried out using Bio-X-Act Long DNA Polymerase (Bioline, London, UK).

### 2.6. Total RNA Extraction and cDNA Synthesis

Total RNA was isolated from *A. thaliana* seedlings using the GeneJet Plant RNA Purification Mini Kit (Thermo Fisher Scientific), following the manufacturer’s instructions.

The purified total RNA was treated with RQ1 RNAse-free DNAse (Promega, Southampton, UK) at 37 °C for one hour and the reaction terminated with RQ1 DNAse Stop Solution by incubation at 65 °C. The DNAse-treated RNA was divided into two halves. One half was subjected to cDNA synthesis using SensiFAST cDNA Synthesis Kit (Bioline). The other half of the DNAse-treated RNA was used as a minus-RT control for genomic DNA contamination. Both the plus RT and the minus RT were treated with RNAse H (New England Biolabs, Hitchin, UK) at 37 °C for 20 min and inactivated at 65 °C for 20 min. All samples were finally diluted 10× in nuclease-free water.

### 2.7. Quantitative Reverse Transcription PCR (qRT-PCR)

The reaction mixture was prepared following the manual by SensiFAST SYBR Hi-ROX Kit (Bioline) in a 96-well plate (Bio-Rad, Watford, UK). *Arabidopsis* genomic DNA solutions of 50, 500, 5000 and 50,000 copies/µL were used to create a standard curve for transcript copy number analysis. A no-template-control (NTC) was prepared with nuclease-free water to replace the cDNA template in order to check for DNA contamination. All qPCR reactions were prepared in triplicate and performed with an ABI PRISM 7000 Sequence Detection System (Applied Biosystems). Transcript copy numbers were normalised against the 18S transcript copies in each sample. The primers used are listed in Table 1.

### 2.8. Biolistic Particle Bombardment

Biolistic bombardments were carried out on 4 cm^2^ onion squares, which were mounted on 1% agar plates to hold them in position to be fired upon and to maintain the cells’ hygrometry. DNA-coated gold particles were vortexed for 1 min and sonicated for 30 s before a 10 µL volume of the particles was deposited onto the micro-projectile. Samples were fired using 1100 psi rupture discs in a PDS-1000/HE particle gun (BioRad) at a distance of 9 cm under a vacuum of −27 inHg. Bombarded onion squares were then kept epidermal side down on the agar plates, which were kept in the dark at 22 °C for at least 24 h before analysis.

### 2.9. Plasmid Construction of AtMC5::RFP

The *AtMC5* cDNA (*At1g79330*) was amplified by PCR, run on an agarose gel and extracted using the Nucleospin Extract II kit (Macherey-Nagel, Dueren, Germany). AtMC5 and pENTR1A were digested with XhoI and EcoRI, and the ligation was performed using the Rapid DNA Ligation Kit (Roche, Basel, Switzerland). The ligation reaction was transformed into an *E. coli* DH5α strain, and transformants were checked by colony PCR. The plasmid was then purified and sequenced. An LR reaction was carried out to transfer the entry vector pENTR1-AtMC5 to the destination vector pH7RWG2 using the Gateway LR Clonase Enzyme Mix II (Invitrogen, Renfrewshire, UK). The AtMC5-RFP construct was then transformed into *E. coli*, checked by colony PCR, purified, and sequenced.

### 2.10. Purification of His-Tagged Recombinant Protein

The plasmid pLIC-AtMC5 was transformed into *Escherichia coli* strain BL21 competent cells (Stratagene, Milton Keynes, UK). A colony was inoculated in 5 mL of LB low-salt medium with the appropriate antibiotic. The pre-culture was grown overnight at 37 °C and 250 rpm and the volume of LB low-salt medium adjusted to achieve an OD of 0.3. The culture was further incubated for 45 min to achieve a final OD of 0.5. The expression of AtMC5 was induced by adding 1 mM of IPTG to the culture, which was then grown at 37 °C and 250 rpm. After 2 h of incubation, the culture was centrifuged at 3000× *g* and 4 °C for 30 min. The cell pellet was frozen in liquid nitrogen and stored at −80 °C for at least 2 days. After 2 days, the cell pellet was resuspended in lysis buffer and incubated for 30 min at room temperature, followed by 5 min of sonication. The soluble protein fraction was obtained by collecting the supernatant after centrifugation at 13,000× *g* and 4 °C for 30 min and purified on a Cobalt/Nickel IMAC resin (ClonTech, Oxford, UK).

### 2.11. Protein Concentration

Pierce 660 nm Protein Assay (Thermo Fisher Scientific) was used for the measurement of all protein concentrations.

### 2.12. SDS-PAGE

Appropriate volumes of protein samples were mixed with 2× Laemmli buffer (100 mM tris-HCL pH 6.8, 4% SDS, 0.2% bromophenol blue, 0.2 M DTT), followed by protein denaturation at 95 °C for 5 min. The samples were separated using Mini-PROTEAN Precast Gels (Bio-Rad) at 150 V for 1 h. The gel was then stained with Instant Blue (Expedeon, Cambridge, UK). The protein marker was the Dual Colour Precision Plus Protein Standard (Bio-Rad).

### 2.13. Western Blot

The proteins on the precast SDS-PAGE gel were transferred to Amersham Hybond P 0.45 µm PVDF (GE Healthcare, Buckinghamshire, UK) at 100 V for 1 h. The membrane was then blocked (3% bovine albumin serum in PBST), probed with primary and secondary antibodies, and washed with PBST using the SNAP i.d Protein Detection System (Millipore, Watford, UK). The primary antibody used for detecting AtMC5 is anti-His (SLS, Nottingham UK), and its secondary antibody is anti-mouse (Amersham Biosciences, Buckinghamshire, UK). All antibodies were diluted 1:3000 in blocking buffer unless otherwise stated. Protein bands were visualised using SuperSignal West Pico and Femto ECL on CL-XPosure films (Thermo Fisher Scientific).

### 2.14. Optimisation of AtMC5 Enzymatic Assay

The AtMC5 activity assays were carried out in a total volume of 100 µL with 5 µL of purified recombinant protein, 50 µM of Boc-GRR-AMC or Boc-VRPR-AMC (Bachem Ltd., St. Helens, UK) and assay buffer (50 mM Tris-HCl, pH 7, 3 mM DTT). The pH and concentrations of CaCl_2_ and sucrose were varied to optimise the activity. Each assay was performed in triplicate in a 96-well plate. The results were measured and analysed using a Fluoroskan Ascent Microtiter Plate Fluorometer (Thermo Labsystems, Philadelphia, PA, USA), where the release of AMC was recorded every 2 min at 37 °C for 30 min using an excitation wavelength of 355 nm and an emission wavelength of 460 nm. The enzymatic activities of the protein were expressed in fluorescence units (FLU) per minute per mg of protein.

### 2.15. Fluorescence Imaging

Leica DM5500 (Leica Microsystems, Wetzlar, Germany) fitted with a Photometrics cascade II 512b EMCCD camera was used for the fluorescence imaging with 40× objectives. Two filter cubes were used: GFP (excitation 470 nm, emission 525 nm) and TX2 (excitation 560 nm, emission 645 nm). All images were processed with SPOT5.5 imaging software.

## 3. Results

### 3.1. Expression of AtMC5::GUS in A. thaliana Tissues

While several metacaspase genes in *A. thaliana* have been studied, there is limited information available on their expression in specific plant tissues. This knowledge is essential to identify which areas of the *A. thaliana* plant require further investigation. To address this, the β-glucuronidase (GUS) gene, a valuable indicator of gene expression, was used to determine whether *AtMC5* expression was specific to particular tissues or organs.

To enable visualisation of the entire seedling, 7-day-old seedlings were used. As depicted in Figure 1, *AtMC5::GUS* was expressed in cotyledons vasculature and roots. The blue staining at the collet was present in WT seedlings and was not specific to *AtMC5::GUS* seedlings. In older 10-day-old seedlings, a faint GUS staining was observed in the stomata of both *AtMC5::GUS* and wild-type plants, representing background in this organ. *AtMC5::GUS* specific expression was found in trichomes and cotyledons (Figure 2), with no substantial difference in blue-stain intensity between 4 h and 16 h. *AtMC5* expression was also visible in the veins of the expanding cotyledon area after 4 h of staining. Upon longer incubation for 16 h, the blue stain appeared to fill the veins towards the petiole. *AtMC5::GUS* expression was not detected in newly opened *A. thaliana* flowers on 5-week plants grown in soil under long-day conditions (Figure 2D).

In addition, blue-stained cells were found at the collet and the quiescent centre in the root apical meristem in both Col-0 WT and *AtMC5::GUS* lines, representing background in these cells (Figure 3). *AtMC5::GUS* specific expression was observed in roots, specifically at the vascular tissues (Figure 3). Along the vasculature, *AtMC5::GUS* was expressed only in a few cells after 4 h of staining, and in more cells after 16 h of staining. Toward the end of the root, the blue stain specific to *AtMC5::GUS* was restricted in a few cells of the lateral root cap. The number of blue root cap cells was increased by longer staining periods.

To investigate whether the expression of *AtMC5* corresponded to local cell death in root cap cells, a Sytox Green staining experiment was conducted. Sytox Green enters dead cells and stains the nucleus, making it an indicator of cell death. As shown in Figure 4, both Col-0 WT and *AtMC5::GUS* lines exhibited green fluorescence signals in a few cells of the root cap. This result is consistent with the pattern of GUS-stained cells observed in Figure 3C.

### 3.2. AtMC5 Responses in Various Stress Inductions

#### 3.2.1. Endoplasmic Reticulum Stress

In addition to identifying where *AtMC5* is expressed in *A. thaliana*, it is also crucial to investigate the factors that induce its expression. Previous research has shown that the presence of tunicamycin, an ER stress inducer, can trigger AtMC5 expression [13]. To validate *AtMC5* expression in this study, a similar approach was employed. Protein-disulfide isomerase (PDI) was used as a marker gene for ER stress [14]. QRT-PCR determined the transcript levels of *PDI* and *AtMC5*, and data were normalised using the *18S rRNA*.

Figure 5A(i) shows that *PDI* expression was significantly upregulated in response to the tunicamycin treatment, indicating that ER stress was induced over time. A low-level expression of *PDI* was observed in the mock sample because its presence is essential for protein folding. Notably, *PDI* expression started to increase at 3 h of tunicamycin treatment and continued to increase steadily until 24 h. In comparison, *AtMC5* expression (Figure 5A(ii)) was nearly absent in the mock sample and at 3 h post-treatment. However, a clear peak in AtMC5 expression was observed at 6 h of tunicamycin treatment. The level of expression immediately declined to a low level after 12 h and remained so at 24 h of treatment, with no significant difference between the two. This suggested that *AtMC5* expression may only be required for a specific short period during early ER stress.

#### 3.2.2. Oxidative Stress

Oxidative stress is caused by the overproduction and accumulation of free radicals and reactive oxygen species (ROS), which are common by-products of plant metabolic processes, such as photosynthesis. When the production of ROS is greater than its systemic detoxification, plant cells can suffer from oxidative stress, leading to programmed cell death [15]. In a study using 3-week-old Col-0 seedlings, methyl viologen induced higher upregulation of *AtMC8* than hydrogen peroxide [6]. To examine whether oxidative stress can modulate *AtMC5* expression, 7-day-old seedlings were used and treated with 10 µM methyl viologen. The overall trend of *AtMC5* transcript expression in response to oxidative stress (Figure 5B) was similar to its expression under ER stress (Figure 5A(ii)), in which the expression level of *AtMC5* had a sudden spike at 6 h after treatment and returned to a low level after 12 and 24 h post-treatment. Notably, the expression of *AtMC5* induced by 10 µM methyl viologen at 6 h was 10-fold less than that induced by treatment with 5 µg/mL tunicamycin at the same time point.

#### 3.2.3. Nitrogen Starvation

Nitrogen is an essential macronutrient that is crucial for plant growth, development and reproduction. In plants, nitrogen is a major component of chlorophyll, highlighting its importance in photosynthesis [16]. To date, no reports have been published on metacaspase expression in response to nitrogen starvation. Therefore, young seedlings grown on nitrogen-rich media were transferred into growing media in the absence of nitrogen to assess the effect of nitrogen starvation on *AtMC5* expression. Figure 5C suggests that *AtMC5* expression was increasingly upregulated throughout the duration of the treatment. After 3 days of treatment, the transcript level in the nitrogen-starved sample was significantly different from the control (mock), albeit at a very low level compared to other stresses.

#### 3.2.4. Leaf Senescence

Senescence is a biological process of ageing in which plants are programmed for cell degeneration that ultimately leads to cell death [17]. Leaf senescence can occur naturally or be induced by placing detached leaves in darkness. During senescence, plants undergo visible phenotypic changes, such as yellowing of leaves. In *Podospora anserine*, the fungi metacaspases *PaMca1* and *PaMca2* have been shown to be involved in senescence-associated cell death [18]. Additionally, an analysis using Genevestigator reported that *A. thaliana AtMC9* was significantly upregulated during senescence [19]. Hence, we investigated whether natural leaf senescence can also affect the expression of *AtMC5*. Senescence-associated gene 12 (*SAG12*) was used as a senescence marker. Figure 5D(i) shows that the senescence marker *SAG12* was upregulated in 3-week-old leaves and significantly increased in 5-week-old leaves, then its expression decreased by nearly half in 8-week-old leaves. This suggests that in our experimental system, leaf senescence occurred in 5- and 8-week-old leaves. In contrast, Figure 5D(ii) reveals that the expression level of *AtMC5* was consistently low in all samples, with non-significant differences up to 5 weeks. A significant increase in the *AtMC5* transcript level was observed to start only at 8 weeks. Therefore, it can be concluded that natural senescence, as detected by SAG12, does not correlate with the expression of *AtMC5*.

### 3.3. Purification of Recombinant AtMC5 Expressed in E. coli

To produce an active AtMC5 via recombinant expression in *E. coli*, the expression vector pLIC, consisting of the *AtMC5* open reading frame (ORF), was transformed into BL21(DE3) competent cells. A 6x-histidine tag was fused to both the N- and C-terminal ends of AtMC5, as shown in Figure 6A, to allow the detection of both ends if the protease were to cleave itself during production for purification. Additionally, an S-tag was present at the N-terminal between a thrombin site and an enterokinase site.

The expression of *AtMC5* in pLIC was controlled by the T7 promoter and induced with isopropyl β-D-1-thiogalactopyranoside (IPTG). AtMC5 accumulated in the cells two hours after IPTG induction at an OD_600_ of 0.5 (Figure 6B, original gel in Appendix A). Subsequently, recombinant AtMC5 was purified under native conditions using immobilised metal affinity chromatography (IMAC) resin charged with cobalt. The histidine tags present at both ends of the recombinant AtMC5 bound to the cobalt ions with high affinity. Samples from each purification step were separated by SDS-PAGE and subjected to western blot analysis, as shown in Figure 6C (original gel in Appendix A). The post-spin supernatant indicated that some of the recombinant protein was in the soluble fraction after cell lysis. It was clear that recombinant AtMC5 was successfully purified in the final elution. The full-length recombinant AtMC5 with a molecular weight of approximately 56 kD was found to be partially cleaved into its subunits, which corresponded to the p10 and p20 subunits of the *A. thaliana* type II metacaspase family [1].

Metacaspases possess a conserved cysteine residue within the p20 subunit domain, which is at position 139 in AtMC5. This residue is critical for the enzyme’s activity, as demonstrated by studies, for example, on AtMC4, AtMC9 and AtMC8, which all showed a loss of enzymatic activity when a single point mutation was introduced at this position [1,6]. To investigate the role of this cysteine residue in AtMC5, a pair of primers was designed to create the amino acid substitution C139A using the QuikChange mutagenesis technique (Figure 6C, original gel in Appendix A). The mutation was confirmed by DNA sequencing. The purified recombinant AtMC5^C139A^ was obtained through the same purification method employed for recombinant AtMC5 WT (Figure 6D, original gel in Appendix A). To evaluate the effect of the C139A mutation on enzymatic activity, 200 µM of synthetic fluorogenic oligopeptide Boc-GRR-AMC was used as a substrate. In comparison to AtMC5 WT, which demonstrated an activity of nearly 500 FLU/min/mg, the mutant AtMC5 ^C139A^ exhibited no activity, as illustrated in Figure 6E. This result demonstrates that the change of amino acid from cysteine to alanine at position 139 (catalytic site) caused the loss of activity of AtMC5.

### 3.4. Enzymatic Activities of AtMC5

Previous studies have shown that the inclusion of sucrose in assay buffers can enhance the stability and activity of purified proteins [20]. Consequently, an experiment was conducted to identify the optimal concentration of sucrose for stimulating the activity of AtMC5. Boc-GRR-AMC was chosen as the primary substrate for the AtMC5 enzymatic assays because it is cleaved by other metacaspases [1,6]. AtMC5 cleaved the synthetic substrate after the arginine residue, liberating the fluorophore, AMC, and the enzymatic activity was recorded as fluorescence units per minute per mg of protein. Figure 7A depicts the AtMC5 activity in response to the addition of sucrose solutions ranging from 0 to 50% into the assay buffer in the absence of added calcium. The differences were not significant, but 15% sucrose yielded the highest peak of activity among the other concentrations tested. Because calcium is required for the activation of other type II metacaspases, such as AtMC4 [21], different concentrations of calcium chloride (CaCl_2_) ranging from 0 to 100 mM were tested in the absence of sucrose. An amount of 10 mM resulted in the highest peak of activity (Figure 7B), and conversely, excessive calcium in the assay buffer appeared to inhibit enzymatic activity, as shown by the lowest activity recorded, with 100 mM CaCl_2_, a 50% decline compared to 0 mM CaCl_2_. The enzymatic activity detected in the absence of added calcium was possibly due to the fact that AtMC5 was already partially activated in *E.coli* or during purification. Once the optimal concentrations of sucrose and CaCl_2_ were established, another assay was conducted using an optimised buffer combining 15% sucrose and 10 mM CaCl_2_. The recombinant AtMC5 demonstrated approximately a six-fold increase in activity compared to the non-optimized assay, as shown in Figure 7C. This result indicated that the enzymatic activity of recombinant AtMC5 can be enhanced by the addition of 15% sucrose and 10 mM CaCl_2_. In order to determine the optimal pH for AtMC5 activity, a wide pH range from 4.0 to 9.0 was tested using various fluorogenic substrates. Figure 7D,E demonstrates a consistent trend, with the enzymatic activity of AtMC5 peaking at pH 7.0 for both GRR and VRPR. To investigate substrate specificity and preference, the same assay was conducted using four different fluorogenic substrates: GRR, VRPR, RR and FR. AtMC5 showed a greater ability to cleave VRPR, resulting in significantly higher enzymatic activity than GRR.

### 3.5. Subcellular Localisation of AtMC5-RFP

An *AtMC5-RFP* T-DNA construct was generated using the expression vector pH7RWG2 via Gateway cloning. To investigate the subcellular localisation of *AtMC5::RFP*, transient expression was carried out using biolistic particle bombardment. *AtMC5::RFP*-coated gold particles were introduced into onion cells, and subcellular localisation was monitored after 24 h of incubation in the dark. Onion cells were selected because they are relatively thin and consist of a single layer of epidermal cells with no photosynthetic pigments. Figure 8 indicates that *AtMC5-RFP* was localised in the nucleus, as indicated by the bright and dense red signal within the nucleus. *AtMC5-RFP* was also observed in the cytosol. The cytosolic localisation of *AtMC5-RFP* was confirmed when the cytosol was retracted inwards from the cell wall upon plasmolysis with 0.5 M mannitol for 30 min.

## 4. Discussion

### 4.1. AtMC5 Is Only Expressed in Specific Cells of A. thaliana without a Strong PCD Association

The *AtMC5::GUS* gene expression under its endogenous promoter was detected in different types of *A. thaliana* tissues, and its strong expression in leaf hairs (trichomes) was an unexpected finding. What could link metacaspase expression and trichome function is currently unknown. AtMC5::GUS expression in the cotyledon veins was notable, although no expression of AtMC5::GUS in the leaves was reported. Another type II metacaspase, *AtMC9*, showed similar results with expression in the expanding cotyledons [22]. Interestingly, *AtMC5::GUS* was also expressed continuously throughout the vascular tissues of the root, similarly to *AtMC9* [23]. *AtMC5::GUS* expression was also detected in a few cells towards the end of the lateral root cap. Bollhöner et al., [24] reported a consistent finding with a cross-section image in the roots of AtMC5::GUS plants. Sytox green staining of roots grown in the same conditions suggested that *AtMC5::GUS* expression in single cells at the end of the lateral root could be associated with root cap cell death, which has been well characterised recently. In conclusion, AtMC5 is only expressed in specific cells of *A. thaliana:* trichomes, cotyledon veins, root vasculature and some cells of the lateral root cap.

### 4.2. AtMC5 Expression Is Inducible by ER Stress and Oxidative Stress

The endoplasmic reticulum (ER) is a protein factory within cells. Any malfunction in its machinery system can impede protein synthesis. Tunicamycin, an inhibitor of protein N-glycosylation, causes protein misfolding and consequently disrupts the ER function. *AtMC5* was found to be highly expressed after 6 h of tunicamycin treatment in continuous light, which is consistent with a study suggesting that *AtMC5* expression was under the control of *NAC089*, a transcription factor induced by ER-stress [13]. Yang et al. [13] showed that *AtMC5* was downstream of the ER membrane-associated transcription factor At*NAC089*, which was linked to plant PCD, as this TF ectopic over-expression caused cell death. It should be noted that we did not observe a cell-death induction in onion cells transiently expressing AtMC5 (data not shown). The expression of *AtMC5* was also found to be induced by oxidative stress induced using methyl viologen. However, at 6 h, there was a sudden spike in *AtMC5* expression, which then returned to low levels at 12 and 24 h post-treatment. *AtMC8* is another metacaspase highly upregulated when treated with methyl viologen and UVC, two treatments inducing PCD [6]; however, for AtMC5, there is a much more time-restricted expression pattern. By contrast, neither nitrogen starvation nor natural senescence induced significant expression of *AtMC5*. Limited nitrogen supply is more commonly associated with autophagy [25]. Although natural senescence has been linked to metacaspases in many different organisms [18,26], *AtMC5* may not have a direct role in this process.

### 4.3. AtMC5 Enzymatic Activity Is Enhanced by Sucrose and Calcium

Recombinant AtMC5 was successfully purified using a His-tag affinity method. Three separate protein fragments were also identified below 37 kD, two of which may correspond to the p10 (small) and p20 (large) subunits of metacaspase, resulting from the processing of AtMC5 in *E. coli* or during the purification process. This finding is consistent with what is known for AtMC4 and AtMC9, where two subunits of 15 and 30 kD amino acids are generated by self-cleavage during activation in a cysteine-dependent manner [1]. However, the processing visible in the inactive AtMC5^C139A^ sample suggested that the observed cleavage can also be generated by other proteases, such as an *E. coli* protease. Like other type II metacaspases, recombinant AtMC5 cleaves substrate after arginine (R). We have also shown that the presence of 10 mM calcium effectively stimulates the activity of AtMC5. Previously, the activation of metacaspase 2 in *Trypanosoma brucei* (TbMCA2) and *A. thaliana* AtMC4 were also reported to be calcium-dependent [21,27]. The stimulation of metacaspase activity by calcium has been suggested to be reflecting a physiological signalling pathway in plant defence, where calcium influx acts as a second messenger in plant cells [3]. Furthermore, the addition of 15% sucrose into the assay buffer with calcium resulted in a sixfold increase in AtMC5 activity. Sucrose has previously been associated with the plant defence response against pathogens [28] and recognised as a signalling molecule in plant innate immunity [29]. Finally, the activity of metacaspases is influenced by pH changes in the cellular environment, and different metacaspases have distinct pH optima. For instance, a study by Vercammen et al. [1], investigated the pH dependence of the *A. thaliana* metacaspases AtMC4 and AtMC9 in vitro and reported that their activity was maximal at pH 7.5 and pH 5.5, respectively. The optimum pH for AtMC5 in this study was found to be at pH 7.0 when tested with at least two different substrates, GRR and VRPR. AtMC5 also had a significantly more prominent cleavage against VRPR as compared to GRR. VRPR was designed as an optimum synthetic substrate for AtMC9 [30].

### 4.4. AtMC5-RFP Is Localised in the Cytosol and Nucleus

The subcellular localisation for *AtMC5-RFP* is cytosolic and not cell wall localised, as validated by the plasmolysis with mannitol in the onion cells expressing AMC5::RFP. This localisation is consistent with the optimal pH at pH 7, demonstrated using the recombinant AtMC5. *AtMC5-RFP* was also observed to be localised in the nucleus in the absence of any stress induction. Two type II metacaspases in *A. thaliana*, *AtMC4* and AtMC9, were described with similarly sub-cellular localisation. AtMC9 was described as localised in both cytosol and nucleus [22]; AtMC4 was shown to translocate from the cytosol to the nucleus under zeocin treatment [12].

## 5. Conclusions

In summary, we have characterised *AtMC5* at the transcript and protein levels. Although metacaspases are often annotated as PCD markers, AtMC5 was found to be induced by ER and oxidative stress but with no strong association with PCD. These findings provide valuable insights for future investigations into the function and mechanism of action of *AtMC5*. The data provided will guide the phenotype characterisation of *atmc5* mutants and support the analysis of protein-protein interaction data.

## Figures and Tables

**Figure 1 biology-12-01155-f001:**
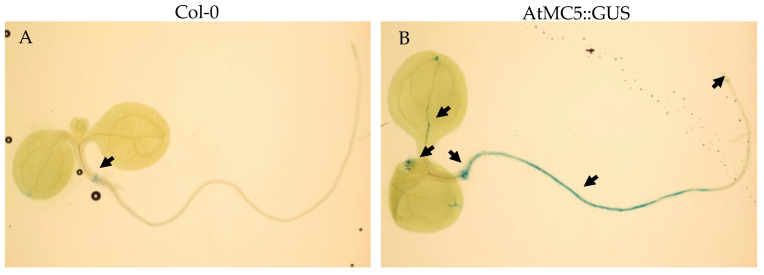
Expression of *AtMC5* in the whole seedlings. Seven-day-old *A. thaliana* seedlings of (**A**) Col-0 WT and (**B**) AtMC5::GUS were subjected to GUS staining with 1 mM X-gluc and in the presence of potassium ferri- and ferrocyanide. Images were taken using a dissecting microscope after 16 h of incubation. Scale bars represent 2 mm. The arrows indicate GUS-stained areas. Images are representative of at least 10 seedlings per condition from 2 independent experiments.

**Figure 2 biology-12-01155-f002:**
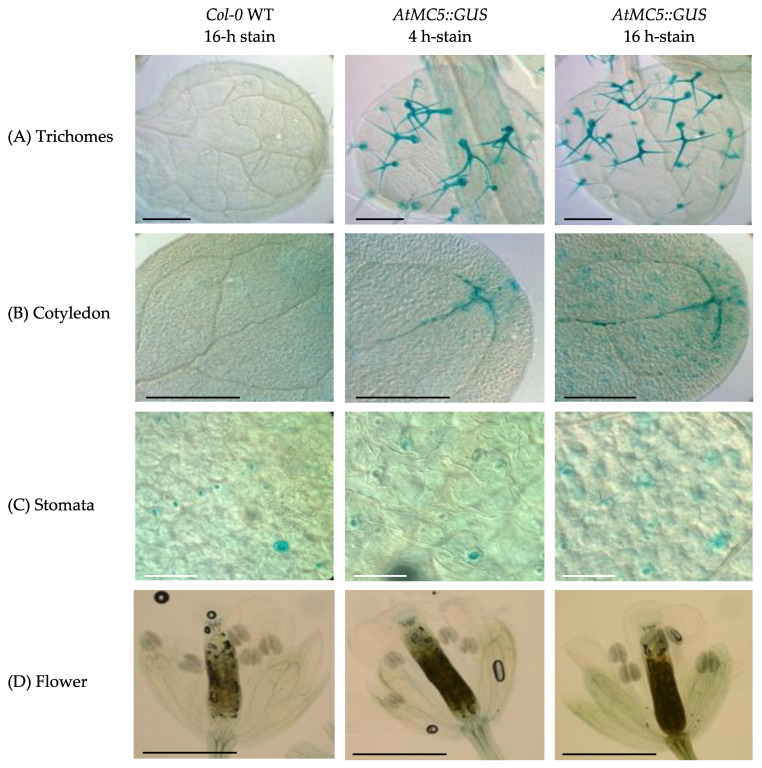
Expression of *AtMC5::GUS* in the upper part of the seedling. Blue staining was observed in (**A**) trichomes, (**B**) cotyledon and (**C**) stomata. There was no staining in (**D**) flowers. Ten-day-old *A. thaliana* seedlings (**A**–**C**) and newly opened flowers from 5-week *A. thaliana* plants (**D**) were subjected to GUS staining with 1 mM X-gluc in the presence of potassium ferri- and ferrocyanide. Incubation in the staining buffer was carried out for 4 and 16 h. Images were taken with a Leica DM5500. The black or white scale bar represents 1 mm. Images are representative of at least 10 samples per condition from 2 independent experiments.

**Figure 3 biology-12-01155-f003:**
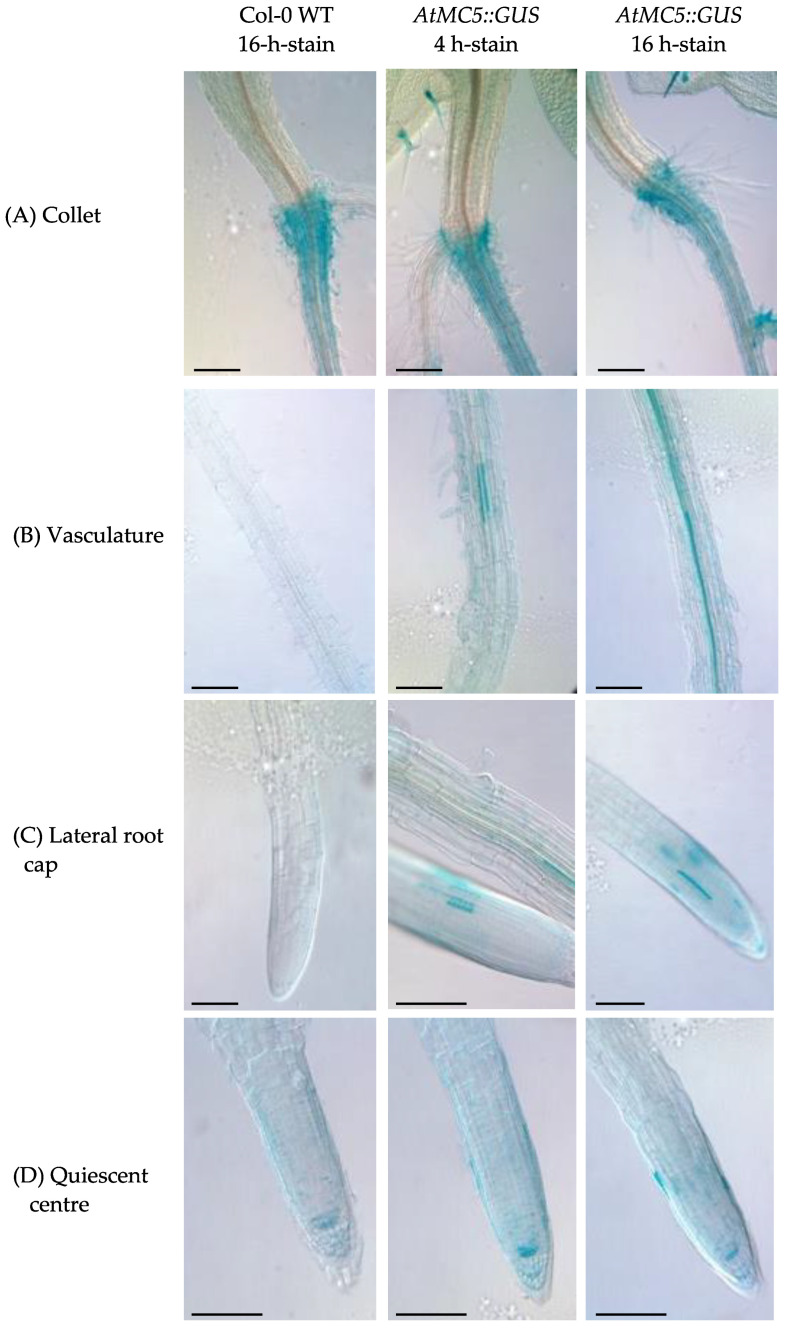
Expression of *AtMC5::GUS* in the *A. thaliana* roots. Blue staining was observed in (**A**) the collet, (**B**) root vasculature, (**C**) lateral root cap and (**D**) quiescent centre. Ten-day old *A. thaliana* seedlings were subjected to GUS staining with 1 mM X-gluc in the presence of potassium ferri- and ferrocyanide. The incubation in the staining buffer was carried out for 4 and 16 h. Images were taken with Leica DM5500. Black scale bars represent 500 µm. Images are representative of at least 10 seedlings per condition from 2 independent experiments.

**Figure 4 biology-12-01155-f004:**
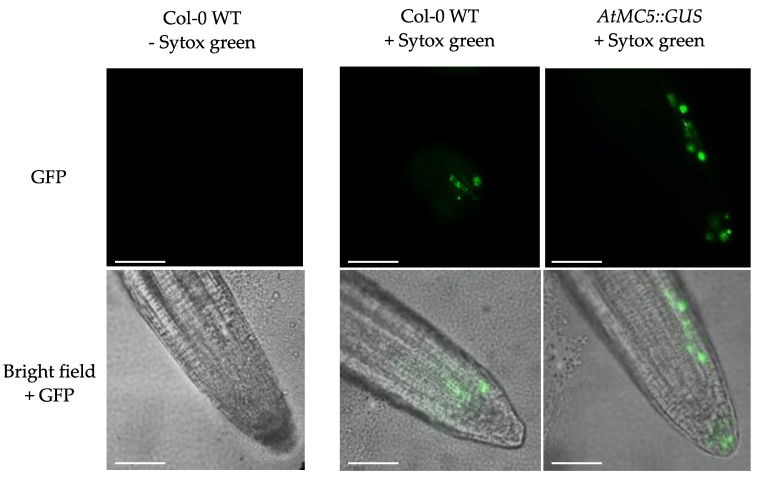
Sytox green staining of *A. thaliana* roots. Ten-day old *A. thaliana* seedlings were incubated with 1 µM sytox green at room temperature for 10 min. Images were taken with a Leica DM5500 with a GFP filter. White scale bars represent 250 µm. Results are representative of three roots per condition.

**Figure 5 biology-12-01155-f005:**
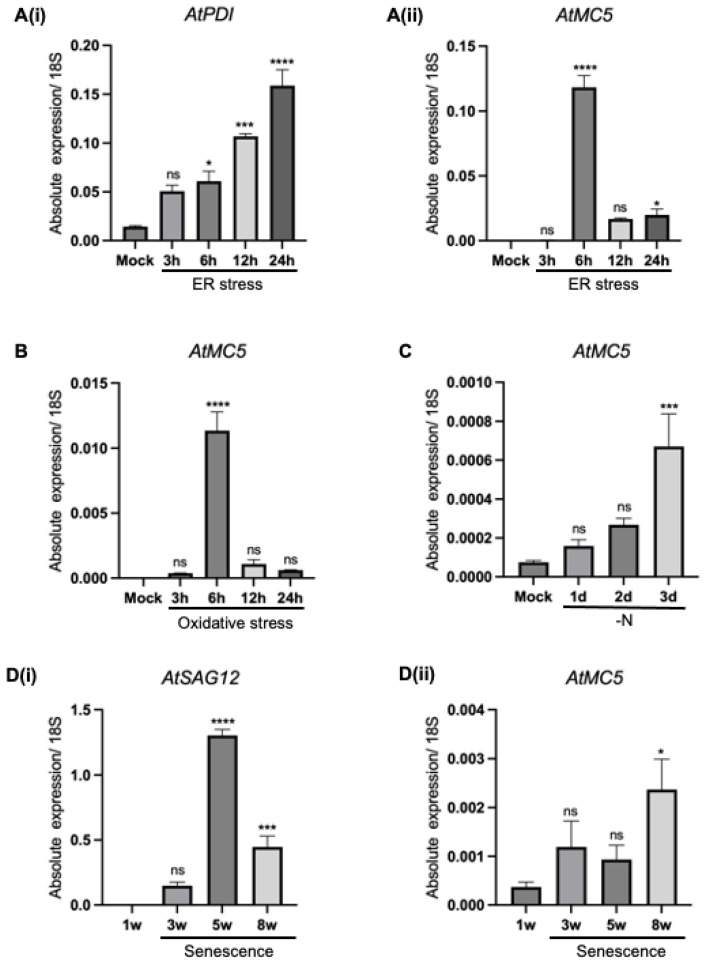
Expression of *AtMC5* in response to various stresses. Seven-day old Col-0 seedlings grown on half-strength MS liquid stress were treated with (**A**) 5 µg/mL tunicamycin for endoplasmic reticulum stress and (**B**) 10 µM methyl viologen for oxidative stress and kept under continuous light at 22 °C. In (**C**) nitrogen starvation treatment, 7-day old seedlings were transferred into media without nitrogen and grown under an 8 h light/16 h dark cycle at 22 °C for 3 days. For (**D**) senescence, leaves were picked from 1-, 3-, 5- and 8-week-old plants. The error bars represent standard deviations of triplicates. The data were analysed using one-way ANOVA with Dunnett’s post-test. **** *p* ≤ 0.0001, *** *p* ≤ 0.001, * *p* ≤ 0.05, ns: not significant.

**Figure 6 biology-12-01155-f006:**
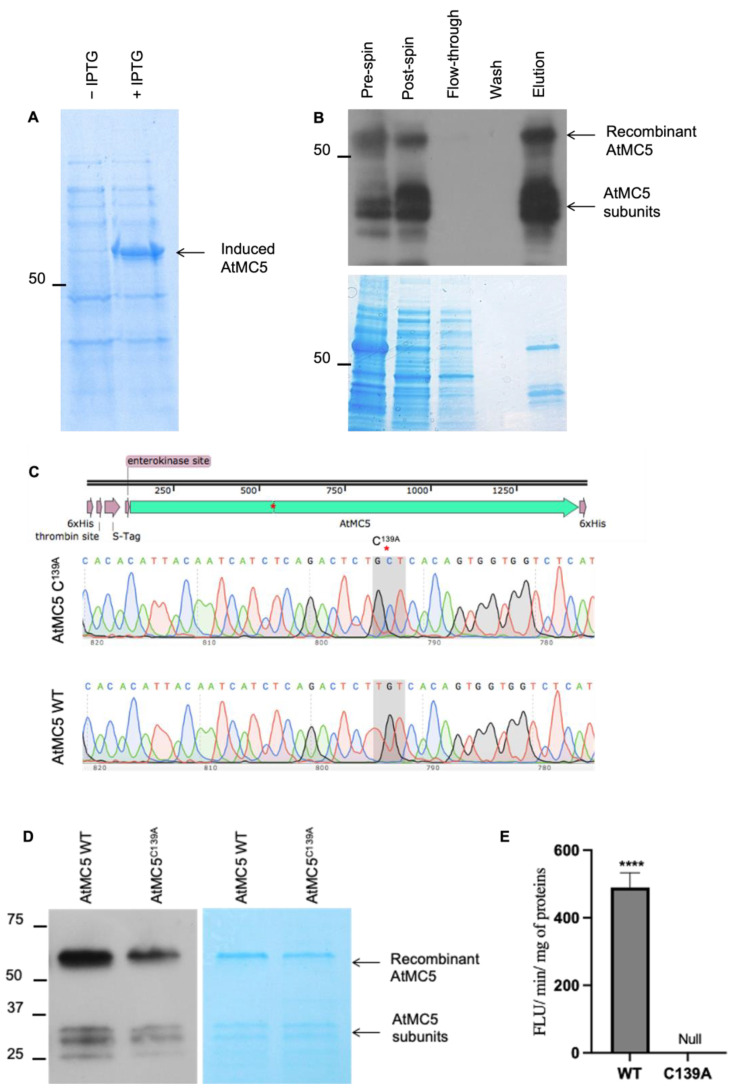
Expression of recombinant AtMC5 wild type and C^139A^ in *E. coli*. (**A**) Crude extract from bacterial cultures containing IPTG (+IPTG) and no IPTG (− IPTG) was separated by SDS-PAGE and stained with Instant Blue. Original gel in Appendix A. (**B**) Recombinant AtMC5 wild type was purified using a His-tag affinity resin. A western blot was run using anti-His (1:3000) as primary antibody and anti-mouse (1:3000) as secondary antibody. The purified fraction was also separated through SDS-PAGE and stained with Instant Blue. Original blot and gel in Appendix A. (**C**) Mutation in the catalytic site of AtMC5 was introduced at position C139A. (**D**) Purification of AtMC5 ^C139A^ as described in B. Original blot in Appendix A. (**E**) Recombinant AtMC5^C139A^ shows null activity in the enzymatic activity against Boc-GRR-AMC in comparison to the WT. The data were analysed with an unpaired *t*-test. The numbers of asterisk above the bars indicate the level of significance compared to the control. **** *p* ≤ 0.0001.

**Figure 7 biology-12-01155-f007:**
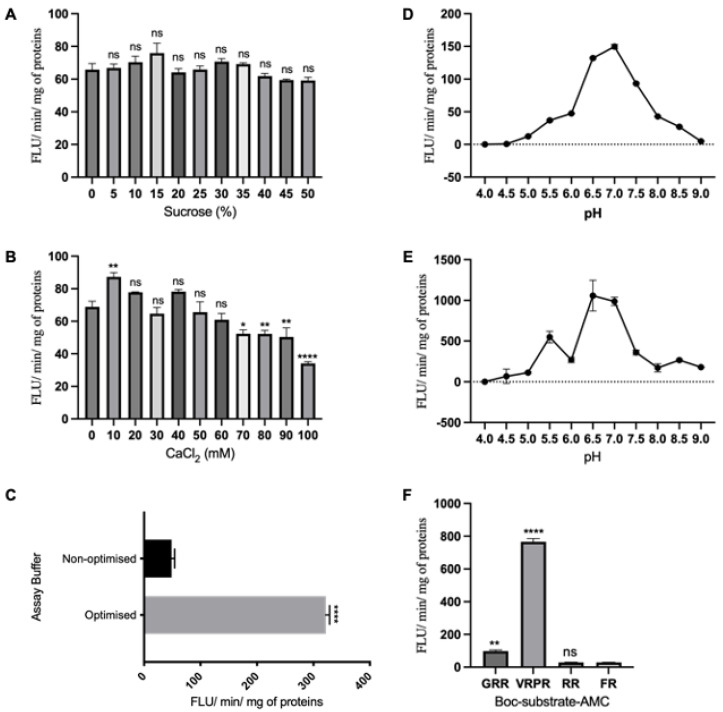
Optimisation of the enzymatic activity of AtMC5. Enzymatic activities of the recombinant AtMC5 against 50 µM of Boc-GRR-AMC with the addition of (**A**) sucrose or (**B**) calcium chloride, CaCl_2_, into the assay buffer, pH 7. (**C**) Comparison between non-optimised (3 mM DTT, 50 mM Tris-HCl, pH 7, no sucrose and no CaCl_2_) and optimised assay buffer containing 15% sucrose and 10 mM CaCl_2_. pH optimisation of AtMC5 activity against different substrates: (**D**) Boc-GRR-AMC and (**E**) Boc-VRPR-AMC. (**F**) Substrate preference by AtMC5. Data from (**A**,**B**,**D**–**F**) were analysed using one-way ANOVA with Dunnett’s post-test, while (**C**) was analysed with unpaired *t*-test. The numbers of asterisk above the bars indicate the level of significance compared to the control. **** *p* ≤ 0.0001, ** *p* ≤ 0.01, * *p* ≤ 0.05, ns: not significant.

**Figure 8 biology-12-01155-f008:**
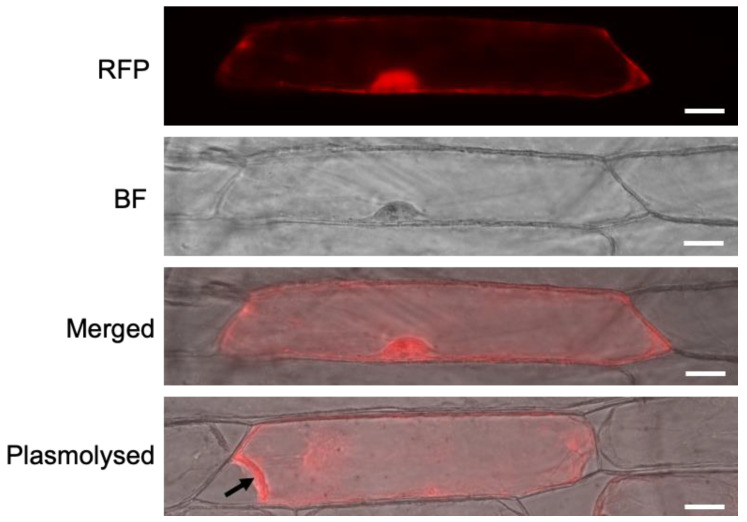
Subcellular localisation of *AtMC5-RFP* in onion cells. AtMC5-RFP was transiently expressed using biolistic particle bombardment, and the images were acquired using a fluorescence microscope 24 h post-transfection. Three panels are shown: RFP filter, brightfield and the merge between both. The bottom image shows a merged image of a different cell that was plasmolysed (cell retraction indicated by the arrow) to confirm the cytosolic localisation of *AtMC5::RFP*. The scale bars in all panels are equal to 50 µM.

**Table 1 biology-12-01155-t001:** List of primers used for qRT-PCR.

Primer	Sequences (5′-3′)
Bip2_F	TCAGCACCAAGTCCGTGTAG
Bip2_R	CTTCACAGGTCCCATGGTCT
PDI_F	TGAGAAATGGAGGGAAGTCG
PDI_R	CAACAACCTCAGTGGCAGAA
18S_29F	GGTCTGTGATGCCCTTAGATGTT
18S_102R	GGCAAGGTGTGAACTCGTTGA
MC5_580F	ATCTAAAGGGATCGCCATTCC
MC5_640R	AGCTTAAACTAGAAGATGGGGCAA
MC5_811F	AAGCTGCAAGAAGGTAAAACTGAAG
MC5_904R	TTAAGCATAAACTAAACGATGACGAAG
MC5v3_F	CTAACAAAGCTGCAAGAAGG
MCFv3_R	GGGAATGATTGGGAAACTAG
ATG5_F	TCTCAACAAGTTGTGCCTGAG
ATG5_R	GTACGAGATGTCATCCCAGGT
GPX8_F	GATGAAGCTTTGTTGCTGAATCG
GPX8_R	TCCAAGTCAATGTCATGGCTCTT
SAG12_F	CAGCTGCGGATGTTGTTG
SAG12_R	CCACTTTCTCCCCATTTTG

## Data Availability

All clones and lines used are available upon written request.

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
