# Peer review of "Characterising the Gene Expression, Enzymatic Activity and Subcellular Localisation of Arabidopsis thaliana Metacaspase 5 (AtMCA-IIb)"

_biology, 2023, doi:10.3390/biology12091155_

Round 1

Reviewer 1 Report

In the manuscript submitted for review entitled: "Characterising the gene expression, enzymatic activity and sub-cellular localisation of A. thaliana metacaspase 5 (AtMCIIb)" the authors Sobri and Gallois initiated a characterization of the A. thaliana metacaspase 5 gene using GUS expression and analysis of expression under various stresses or developmental stages, and enzyme activity with purified recombinant AtMC5 and sub-cellular localization with an AtMC5-RFP fusion protein.

The very brief introduction to the various plant metacaspases and more particularly those of class II is clear. Contrary to the clear introduction, the material and method part seems to lack a lot of information and is sometime confusing. Therefore, we recommend the following changes:

The result of the figure 4 on the staining with sytox green does not appear in the material and method. Analysis of the AtMC5 expression during oxidative stress is also not included in the material and method. Please add these missing informations.

Additionally, regarding the Gus histochemical staining, the authors inform the use of 2 mM of the substrate 5-bromo-4-chloro-3-indolyl β-D-glucuronide (X-gluc). However, in the results, figure 2 and 3, authors tell us about 1 mM of X-gluc. Please clarify.

The method for analyzing expression during senescence also does not appear in the method and the few information of the method in the result part seem to be confusing. Indeed, the authors tell us that they have analyzed the senescence on 1, 3, 5 and 8 week-old plants and sometimes on 1, 3, 5 and  8 week-old leaves. Since material and methods only mention culture in a petri dish, it is difficult to imagine a good plant growth for 8 weeks in these conditions. A lack of space and macro- and micronutrient deficiencies may have affected the analysis of natural senescence. Moreover, if all senescent leaves were harvested at these different plant age (1, 3, 5 and 8 week-old plants), the pools of leaves were harvested at different stages of senescence and therefore do not represent senescence kinetics. The result obtained without explanation of the methods cannot affirm the absence of induction of AtMC5 during leaf senescence. Moreover, it would be preferable for the authors to speak of "leaf senescence" than of “senescence”, because the actors in these processes can vary depending on the organ.

To continue with the results section, the authors attempt to analyze a spatial activity of the AtMC5 promoter with a GUS construct. They made descriptions but the photos do not allow us to verify what is described. For figure 1, the authors described the expression of AtMC5::GUS in the trichome in the true leaves but the image corresponds mainly to the cotyledon, and the true small true leaves are very hard to discern. Please clarify in the text.

In addition, in figure 2 the authors describe the presence of GUS stain in the tracheary element and leaf xylem in the true leave but it’s difficult without cross section to affirm if it is xylem or phloem or the entire stele… Same comment for the assertion of the presence of GUS stain in the root xylem in the figure 3.

Contrary to the results which do not allow this reviewer to have great confidence in the results on the activity of the promoter of AtMC5 in the xylem of leaves and roots and the absence of expression of AtMC5 during senescence, the results on the enzymatic activity and the subcellular localization of AtMC5 are not to be reproached and call upon interesting materials and methods. These results provide an interesting basis for helping the scientific community characterise matacaspases, and AtMC5 in particular. However, their presentation lacks a bit of rigor. There is one point related to Figure 7 though: This reviewer is surprised to see such high activity of MC5 at 0 mM Ca2+ concentration. That must mean there is plenty of Ca2+ already present in the extraction buffer. This can be tested with the addition of EGTA to chelate potentially contaminating Ca2+. Ca2+ is required not only for activation (autocatalytic cleavage), but also for sustained activity. See Štrancar et al 2022, DOI: 10.1016/j.isci.2022.105247

Other remarks :

Abstract – according to the new metacasapse nomenclature agreed upon by an exhaustive list of experts in the field, including the corresponding author, AtMC5 should be named AtMCA-IIb. Please change. It is fine to stick to AtMC5 throughout the text, but on first mention the double name should be specified, including for the other metacaspases mentioned in the introduction. See Minina et al. 2020 (doi: 10.1016/j.molcel.2019.12.020)

Figure 1 - ´xylem of the cotyledons´… This is impossible to discern without transversal sections through the cotyledon – vasculature or vascular tissue is the more appropriate term. Same goes for Figure 2.

Figure 2 – Impossible to have flowers on a 10-day old seedling! Please change the results text describing this figure (´In older 10-day-old seedlings, AtMC5::GUS expression was found in trichomes, leaf xylem, stomata, and flowers´). The GUS stain in stomates is unspecific as it also appears in the Col-0 seedlings. Same for the flowers – this reviewer cannot discern where is the GUS stain in the flowers different to the Col-0 seedlings.

The results section is contradictory for Figure 2 and 3. The paragraphs start with ´MC5 expression in x,y, and z,´ only later to say that some of the GUS stain is also visible in Col-0 plants – there is no expression in stomata, flowers, and quiescent center. Please make this clear. The authors did well to include a Col-0 control!

Discussion This sentence contradicts itself: ´AtMC5::GUS expression in the xylem of cotyledons and true leaves was notable, although expression of AtMC5::GUS in the leaves was reported.´ Please clarify.

In section 2.7 of M&M «which were kept in the dark at 22°C for at least 24 hours before analysis." », remove the quotation mark.

Many references are missing in the list of references : Stael al., 2013, Coll et al. 2014, Wrzaczek et al. 2015, Sundström et al. 2009, Hander et al. 2019, Shen et al. 2019, Huysmans et al. 2018, Stael et al. 2022…

Many references are in the reference list and not in the text: Bush,1995 , Hülskamp, 2004,  Lecourieux et al., 2006,  Lee et al., 2009,  Safadi et al., 1997…

Add the “h” at Belenghi et al, 2018: “VRPR was designed as an optimum synthetic substrate for AtMC9 (Belengi et al. 2008).”

(Watanabe & Lam, 2011), you have two references for the same authors at the same date in the reference list differentiated by an “a” or a “b”, please annotate with an “a” or a “b” also in the text.

Sometimes et al., is in italics and sometime not.

Replace some “et al.” by “et al.,” .

Reviewer 2 Report

manuscript is generally clear. It's publishable with revisions.

All abbreviations should be spelled out in full

AtMC5 Is Only Expressed in Specific Cells, it would be better to summarize which specific cells are present, please revise "In conclusion, there are many cell types that do not express AtMC5 constitutively. constitutively" to "AtMC5 Is Only Expressed in …Cells”.

The Simple Summary needs to include the specific highlights of this manuscript, rather than a generalization significance of your research on AtMC5. Of course, if your Journal requires it, then you should comply with your Journal's requirements.

A. thaliana in the title suggests the full name Arabidopsis thaliana.

If the authors do an experiment with the material in Fig. 1 to illustrate the distribution of AtMC5 in PCD or aged leaves, then it can be concluded that "We found no evidence of a link between AtMC5 and PCD". The results of Figure 5D show that there is still a significant difference in the expression of AtMC5 at 8 week compared to 1 week, the amount detected values is very small, maybe the gene has a faster turnover rate? This could be confirmed by further experiments in the future.

There are some minor errors that need to be carefully checked and corrected, such as 4 cm2 in “2.7. Biolistic Particle Bombardment”, where the 2 needs to be superscripted; and a space between 0.5 and M in the last sentence of 0.5M in “3.5. Subcellular Localization of AtMC5-RFP”.
